# Diagonal Batching Unlocks Parallelism in Recurrent Memory Transformers for Long Contexts

## Abstract

Long-context inference with Transformers is constrained by quadratic attention and linear memory growth. Many linear-time alternatives require pretraining from scratch, whereas Recurrent Memory Transformers (RMTs) convert pretrained models into segment-recurrent variants via finetuning without modifying the original model architecture. However, their sequential memory updates underutilize GPUs. We show that RMT-style architectures with *layer-level* memory (PRMTs) (e.g., ARMT) can be among the most latency-efficient linear approaches when scheduled properly. We introduce *Diagonal Batching*, a compute-reordering scheme that preserves exact recurrence while exposing inter-step parallelism by executing "diagonals" concurrently with grouped layers. On LLaMA (1B/3B/8B) up to 131,072 tokens on A100/H100, Diagonal Batching achieves up to $3.3\times$ lower latency than full-attention inference and $1.8\times$ over a sequential ARMT baseline, with *no custom CUDA kernels*. With the right scheduling, PRMTs achieve linear scaling with context length and stand out as competitive, scalable architectures among linear recurrent models.

## 1 Introduction

Transformer-based language models have not only revolutionized natural language processing (NLP) (Vaswani et al., 2017; Devlin et al., 2019; Radford et al., 2019), but also catalyzed the development of intelligent agents that can solve complex, multi-step problems in various domains by scaling up to large language models (LLMs) (OpenAI, 2023; Reid et al., 2024; Dubey et al., 2024). However, these transformer-based models have quadratic time complexity and a linear memory footprint with respect to the length of the input sequence. Consequently, real-world applications are limited by the context window size of standard transformers that can fit within hardware constraints.

From an engineering perspective, numerous optimizations have been proposed to improve attention efficiency and manage GPU memory more effectively. Optimized attention kernels, such as FlashAttention (Dao et al., 2022; Dao, 2024) and the xFormers library (Lefaudeux et al., 2022), focus on reducing memory access overhead and maximizing throughput. Memory-saving attention modifications like Multi-Query Attention (MQA) (Shazeer, 2019), Grouped Query Attention (GQA) (Ainslie et al., 2023), and Multi-head Latent Attention (MLA) (Liu et al., 2024a) lower GPU RAM usage by sharing and optimizing KV-cache. For distributed long-context training, methods like Ring Attention (Liu et al., 2024b) and Microsoft DeepSpeed's Ulysses (Jacobs et al., 2023) partition sequence data across multiple devices to scale beyond single-GPU memory limits.

Along with these engineering optimizations, alternative architectures to the standard Transformer have been explored. Recently, state-space and linear recurrent models, such as S4 (Gu et al., 2021), RWKV (Peng et al., 2023), RetNet (Sun et al., 2023), and Mamba (Gu & Dao, 2023; Dao & Gu, 2024) have replaced the softmax attention with alternative read-write operations. These models offer efficient parallel training, like transformers, and require constant memory during inference, like RNNs. However, these approaches often suffer from reduced memory capacity (Jelassi et al., 2024) and decreased accuracy in read-write operations (Rodkin et al., 2024). Furthermore, both state-space models and Transformers face theoretical limits, such as the $TC^0$ complexity bound on the class of

functions computable in a single forward pass (Merrill et al., 2024; Strobl et al., 2024), constraining their expressivity despite massive parallelism.

Memory-augmented models (Weston et al., 2015; Sukhbaatar et al., 2015), especially memory-augmented transformers with segment-level recurrence (Dai et al., 2019; Rae et al., 2020; Bulatov et al., 2022; Hutchins et al., 2022) offer an alternative approach by compressing history into fixed-size memory states and propagating them across segments. In Recurrent Memory Transformers (RMT) (Bulatov et al., 2022), special memory tokens carry state between segments, and each Transformer block acts as a recurrent cell. This approach reduces inference complexity to linear time and constant memory, supporting arbitrarily long contexts (Bulatov et al., 2024). However, the recurrent nature of RMT makes it not fully parallelizable; all subsequent layers have recurrent dependencies, and all segments must be processed sequentially.

Parallel Recurrent Memory Transformers (PRMTs) (Rodkin et al., 2024) are a broader class of architectures in which each layer maintains its own memory state. PRMTs localize recurrence within layers and eliminate all inter-layer memory flow. The Associative Recurrent Memory Transformer (ARMT) (Rodkin et al., 2024) belongs to this family and demonstrates exceptional scalability. It maintains high quality on sequences of up to 50 million tokens, which is far beyond the capacity of RMT and Mamba (Rodkin et al., 2024; Kuratov et al., 2024). Models such as RWKV, Mamba, and other linear-recurrent architectures can also be considered members of the PRMT family due to their layer-level memory design.

PRMTs are asymptotically linear, yet they run sequentially over segments, which underutilizes GPUs for single, long input requests. Naive micro-batching and pipelining are not helpful because they require sophisticated batching over very long input sequences. This leads to unpredictable SLAs and even higher latencies, due to the co-execution of many large-context requests at the same time. Moreover, for pipelining, kernels operate on small inputs (segment sizes typically under 1024), leading to small kernels that are unable to utilize the GPU without micro-batching.

In this work, we introduce *Diagonal Batching*, a scheduling scheme that unlocks inter-segment parallelism in PRMTs inference without altering their exact recurrence. By reorganizing the 2D grid of layer and segment computations into independent "diagonals", our method enables concurrent execution of up to N_Layers operations per GPU kernel launch, eliminating the need to use complex pipelining or micro-batching at all, which greatly simplifies the complexity of large context deployments. Diagonal Batching fully encapsulates transformer block computations across segments, thus *eliminating the layer- and segment-level synchronization barriers* present in previous RMT implementations. Diagonal Batching does not require writing custom CUDA kernels to achieve efficiency.

We implement Diagonal Batching in the ARMT framework and evaluate its performance on a LLaMA-1B, 3B, and 8B models with sequence lengths up to 131,072 tokens on an NVIDIA A100/H100 GPUs. Our experiments demonstrate a $3.3\times$ speedup over standard full-attention inference and a $1.8\times$ improvement relative to a sequential ARMT baseline for 1B models. We show that RMT-style architectures with layer-level memory (PRMTs), such as ARMT, are among the most latency-efficient linear approaches for long-context inference when scheduled properly (via Diagonal Batching).

Our contributions are:

1. We identify execution *scheduling*, rather than algorithmic complexity, as the primary utilization bottleneck for RMT-style linear recurrent models, especially on small and medium segment sizes.

2. We show that ARMT linear transformer become highly latency-efficient when scheduled right with *Diagonal Batching*, a simple, kernel-agnostic grouping schedule that preserves exact recurrence and exposes up to $N_{\text{layers}}$ inter-step parallelism, yielding near-linear latency scaling without custom kernels.

3. We empirically show that, ARMT with Diagonal Batching exhibits the best *latency scaling with context length* among the linear-recurrent baselines we tested (Mamba, RWKV); on LLaMA-1B at 131,072 tokens it achieves $3.3\times$ lower latency than full attention and $1.8\times$ over a sequential ARMT baseline, with $\approx 1\%$ relative logit drift, comparable to the drift observed between SDPA and FlashAttention.

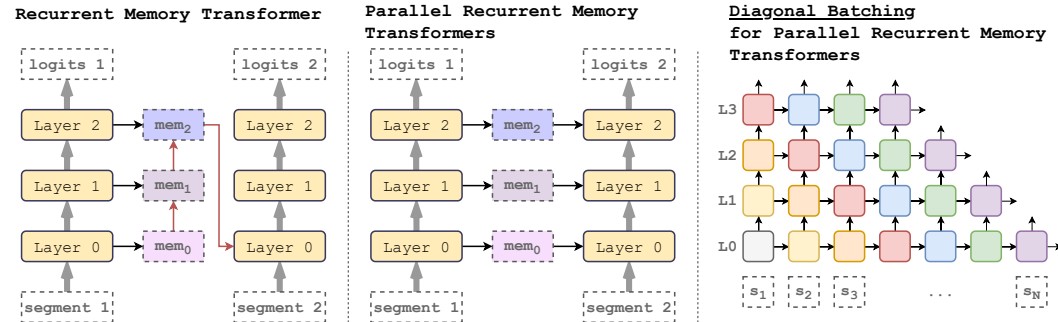

Figure 1: **Unlocking Parallelism in Recurrent Memory Transformers (RMT) with Diagonal Batching. Left:** RMT splits long sequences and processes segments sequentially. Each layer updates a memory ($\text{mem}_0$, $\text{mem}_1$, ...) and the final-layer memory is fed as input to the next segment; red arrows show the recurrent dependencies that force strictly sequential execution. **Center:** Parallel RMT (layer-level memory): each layer passes its own state to the same layer in the next segment, removing inter-layer dependencies but retaining per-layer segment recurrence. **Right:** Diagonal Batching rearranges the 2D grid of layers (rows) and segments (columns) into independent "diagonals" (same colored blocks). This allows all operations on one diagonal (up to N_Layers) to execute concurrently on the GPU, thus eliminating the sequential bottleneck while preserving all layer-level recurrence.

## 2 BACKGROUND

### 2.1 RECURRENT MEMORY TRANSFORMERS

**Recurrent Memory Transformer (RMT)** extends standard Transformer architectures by introducing segment-level recurrence (Figure 1, left). Specifically, the hidden representations corresponding to a segment $s$ are conditioned on a recurrent state $M$—referred to as the *memory*—propagated from the previous segment $s - 1$.

In the original RMT formulation, the memory state is implemented as a sequence of input embeddings. The memory update mechanism can be formally expressed as:

$$[\_, \_, M_s] = \text{Transformer}([M_{s-1}, H_{s-1}, M_{s-1}]), \tag{1}$$

where $M_s$ denotes the memory state associated with segment $s$, and $H_{s-1}$ represents the input embeddings from segment $s - 1$. The square brackets indicate concatenation of the input sequences.

**Associative Recurrent Memory Transformer (ARMT)** introduces a parallel memory mechanism designed to support a hierarchical memory structure. Unlike the original RMT, ARMT maintains distinct memory states across different layers. This design facilitates a more expressive memory representation by allowing each layer to store and update its own memory.

The memory update rule in ARMT is formulated as follows:

$$[\_, M_s^l] = \text{TransformerLayer}(\text{AssociativeLayer}([H_{s-1}^{l-1}, M_s^{l-1}])) \tag{2}$$

$$k_i, v_i = W_K m_i, W_V m_i; \quad \beta_i = \sigma(W_\beta m_i); \quad A_0^l = \vec{0}; \quad z_0^l = \vec{0}; \tag{3}$$

$$\overline{v}_i = \frac{A_{s-1}^l \phi(k_i)}{(z_{s-1})^T \phi(k_i)}; \quad \gamma_i = 1 - \frac{(z_{s-1})^T \phi(k_i)}{\|\phi(k_i)\|^2}; \tag{4}$$

$$A_s^l = A_{s-1}^l + \sum_i \beta_i(v_i - \overline{v}_i) \otimes \phi(k_i); \quad z_s^l = z_{s-1}^l + \sum_i \gamma_i \phi(k_i). \tag{5}$$

$$\text{AssociativeLayer}(x_i) = \frac{A_{s-1}^l \phi(W_Q x_i)}{(z_{s-1}^l)^T \phi(W_Q x_i)}, \tag{6}$$

where $m_i$ is the vector from $M_s^l$, $A_s^l \in \mathbb{R}^{d_{\text{model}} \times 6d_{\text{mem}}}$, $z_s^l \in \mathbb{R}^{6d_{\text{mem}}}$, $\phi$ is the untrained nonlinearity DPFP-3 (Schlag et al., 2021), $x_i$. is the vector from $[H_{s-1}^{l-1}, M_s^{l-1}]$.

This mechanism essentially implements quasi-linear attention with a delta rule for segment-level recurrence.

## 2.2 LAYER-LEVEL RECURRENT MODELS

We call a model *layer-level recurrent* if, at time step $t$ and layer $\ell$, the computation depends only on $(t, \ell - 1)$ and $(t - 1, \ell)$ in the layer-time grid. The index $t$ may denote either *tokens* or *segments* (chunks of tokens). We use *Parallel Recurrent Memory Transformers* (PRMTs; Figure 1, center) as a broad label for architectures that satisfy this dependency at either granularity. This class includes ARMT (Rodkin et al., 2024), RWKV (Peng et al., 2023), Mamba (Gu & Dao, 2023; Dao & Gu, 2024), and other linear-recurrent models (Yang et al., 2024).

Per-layer memory enables scheduling policies that exploit parallelism across segments. *Diagonal Batching* targets such layer-level recurrent architectures: it preserves the above dependency while enabling parallel execution across segments. By contrast, RMT (Bulatov et al., 2022) introduces an additional dependency on the previous step's *final* layer; when the step is a segment $s$, output of $(s, \ell)$ also depends on $(s - 1, L)$ (Figure 1, left), which prevents diagonal scheduling.

## 2.3 EXISTING INFERENCE OPTIMIZATIONS TECHNIQUES FOR TRANSFORMER MODELS

Numerous techniques are proposed to speed up the inference of transformer models, including FlashAttention (Dao et al., 2022; Dao, 2024), speculative decoding (Xia et al., 2023), quantization (Frantar et al., 2022; Lin et al., 2024), among others. Practical methods should remain compatible with these optimizations. Diagonal Batching is orthogonal to these methods and integrates with them seamlessly, e.g., it can employ FlashAttention within segments computation and to compute attention efficiently.

**Hardware utilization.** Effectiveness of individual operations is often analyzed via the roofline model, which characterizes the performance limits of hardware based on computational intensity and memory bandwidth (Williams et al., 2009). Transformer architecture mostly consists of matrix multiplication - a compute bound operation. Matrix multiplication's computational intensity does not depend on batch size. However, the total achievable floating-point operations per second (FLOPS) improves significantly, as larger batch sizes enable better parallel workload distribution across GPU cores, optimizing hardware utilization (Dao et al., 2022).

Despite these benefits, a large batch size introduces significant memory demand. It mostly comes from intermediate activation computations and storing output logits, which scales linearly with batch size and sequence length. This limits practical usage of batching, as large language transformers often use almost all available GPU memory.

## 3 DIAGONAL BATCHING METHOD

### 3.1 INTUITION AND DEPENDENCY GRAPH

In the naive approach, we must perform many forward operations (`n_segments × n_layers`) using inputs of shape (`segment_size, hidden_size`). In PRMTs, each (`segment, layer`) pair only depends on the preceding pairs: (`segment, layer-1`) and (`segment-1, layer`).

Given this dependency, all pairs where `segment + layer = i` can be computed in parallel during the $i$-th iteration. Each iteration can be visualized as a diagonal in the forward-pass computation graph, as shown in Figure 1, right.

If the execution is not compute-bound, this diagonal execution approach can yield a significant speedup for PRMT models.

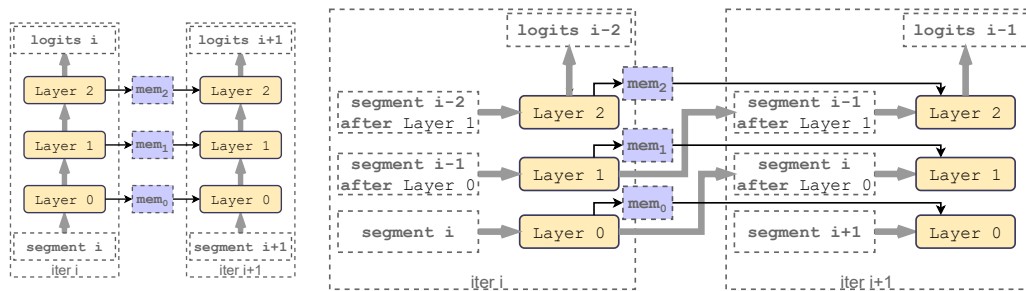

(a) Baseline compute scheme.

(b) Diagonal Batching: grouped compute scheme.

Figure 2: Baseline compute schedule in PRMTs leads to n_layers x n_segments sequential operations. Diagonal Batching reduces this value to n_layers + n_segments by grouped computations.

---

**Algorithm 1** GROUPED ARMT EXECUTION (DIAGONAL BATCHING)

---

**Require:** input sequence $\mathcal{I}$, number of layers $L$, grouped layer $\mathcal{G}$
1:  ZEROGROUPEDMEMORY($\mathcal{M}$)
2:  $segments \leftarrow$ SEGMENT($\mathcal{G}, \mathcal{I}$)                    ▷ token ids to segments with memory tokens
3:  $GInput \leftarrow []$, $Out \leftarrow []$
4:  **for** $i = 0$ **to** $L + |segments| - 1$ **do**
5:      **if** $i < |segments|$ **then**
6:          **prepend** $segments[i]$ to $GInput$                    ▷ ingest new segment
7:      **end if**
8:      $X \leftarrow$ STACK($GInput$)
9:      **if** $i > 0$ **then**
10:         $X_{0:|X|-1} \leftarrow$ ASSOCIATE($\mathcal{G}, X_{0:|X|-1}$)     ▷ memory association operation between consecutive segments
11:     **end if**
12:     $Y \leftarrow$ GROUPEDFORWARD($\mathcal{G}, X$)                    ▷ multi-layer grouped call
13:     UPDATEMEM($\mathcal{G}, Y_{:,-num\_mem\_tokens:}$)                ▷ memory update for next segment
14:     $GInput \leftarrow$ list of segments in $Y$
15:     **if** $i \geq L - 1$ **then**
16:         $O \leftarrow GInput$.POPLAST                    ▷ segment went through all layers
17:         **append** $O$ to $Out$
18:     **end if**
19: **end for**
20: **return** CONCAT($Out$)                                     ▷ final logits

---

## 3.2 BATCHING

Simplified description of the algorithm is given for ARMT in Algorithm 1. For other Parallel RMTs, the algorithm is the same, but without memory association and update operations.

**Lemma 3.1.** *Diagonal Batching completes the DAG in the minimum possible number of groups, $N_{\text{segments}} + N_{\text{layers}} - 1$, and schedules each node $(i, j)$ in its earliest feasible group $i + j$.*

*Proof.* Topologically sort the DAG by the key $(i, j)$ with root being $(0, 0)$. In this ordering, each node $(i, j)$ appears at level $i + j$, which is therefore the earliest group it can occupy, and the longest path has length $N_{\text{segment}} + N_{\text{layers}} - 1$ vertices. Hence, any schedule needs at least $N_{\text{segment}} + N_{\text{layers}} - 1$ groups. Diagonal batching uses precisely those levels as its groups, achieving both bounds. $\square$

## 3.3 IMPLEMENTATION DETAILS

To efficiently implement grouped layer computations, we modify the base model architecture. All layers are replaced with a single grouped layer, as shown in Figure 2. Using the initial layer of the model as the basis, we implement the following adjustments: (1) Replace the linear layers with a `GroupedMatmul` operation. The weights and biases are constructed by stacking those from the original layers. (2) Layer normalization weights are also replaced by stacking parameters across all

layers. Additionally, the forward pass is adapted to ensure correct broadcasting behavior. (3) All other operations remain unchanged. However, they operate as though they handle significantly larger batch sizes, contributing to parallel execution.

For the grouped matrix multiplication, we utilize the `GroupedGEMM` function from the CUTLASS library with a minor optimization: the output tensor is pre-allocated as a single large tensor, which is subsequently partitioned into individual submatrices without additional overhead.

**Difference from pipelining.** Diagonal Batching is a scheduling-and-layer-grouping method, not pipeline parallelism. Unlike pipelines, it (1) uses a single control thread—no multi-thread/multi-stream coordination or intrusive graph rewrites; (2) runs larger kernels instead of many small ones, improving GPU utilization (see Figures 6 and 7) and avoiding CPU small-matrix special-casing (Yang et al., 2021); and (3) requires no micro-batch overlap to hide bubbles as in pipelined systems (Huang et al., 2019; Qi et al., 2023), yet achieves high utilization for single-request inference with a constant-memory pattern, simplifying fleet deployment.

## 4 EXPERIMENTS

In the experiment section, we address two main questions regarding the Diagonal Batching method: How much speedup we can get compared to the naive ARMT implementation in single request inferences? How ARMT with Diagonal Batching scales compared to other linear recurrent models (Mamba, RWKV) and to full-attention models (LLaMA)?

We start by showing that efficiency grows for individual bottleneck operations inside network - linear layers and attention. Then, we show the resulting scaling for the transformer models with ARMT of different sizes. We conducted all experiments with the models from the LLaMA-3 family (Grattafiori et al., 2024).

### 4.1 INFERENCE SCALING

The performance increase for individual operations directly translates into overall model speedup. We evaluate this effect on LLaMA ARMT models of varying sizes—160M (Table 9), 1B (Table 1), 3B (Table 7), and 8B (Table 8).

Across all model sizes and batch configurations, our implementation consistently achieves substantial speedups over the default ARMT implementation. Gains are particularly pronounced for smaller segment sizes. This is because, with larger matrix multiplications, hardware utilization is already near peak FLOPS, leaving less room for group scaling.

A key implication of these results is that researchers can prioritize quality-driven choices for segment size without being overly constrained by performance. Diagonal Batching decouples performance from segment size, allowing better flexibility in architectural decisions.

### 4.2 SCALING BY MODEL FAMILY

We show how the different architecture families scale with input sequence length across different model parameter sizes in Figure 3. Scaling for wider model classes includes measurement for GPT and is shown in Appendix Figure 12. For model sizes under 0.5B, efficiency increase is very significant, so the 3B model under Diagonal Batching performs similarly to the 0.4B model before optimization. For bigger models, the gap is smaller, but allows to outperform the base non-linear model starting from 32k context.

### 4.3 COMPARISON WITH OTHER LINEAR TRANSFORMERS

Comparison with other models shown in Figure 4. More extensive comparison shown in Appendix Figure 11 and Figure 10. Before our optimization, ARMT was slower on many configurations than Mamba, RWKV, and sometimes even a quadratic-complexity transformer. With the Diagonal Batching algorithm, ARMT outperforms other linear transformers with most configurations, providing a cost cut compared to non-linear transformers. For a fair comparison, we use the most efficient

Table 1: Diagonal Batching speeds up the execution for longer sequences — from 1.1× to 2.7× compared to base ARMT at 131072 sequence length. Execution time comparison (in seconds) and relative speedups across different sequence lengths compared to LLama-3.2-1B-ARMT. Configuration format: (segment_size, memory_tokens). Measured on Nvidia A100 GPU.

| Method | Sequence Length | | | | | |
|---|---|---|---|---|---|---|
| | 4096 | 8192 | 16384 | 32768 | 65536 | 131072 |
| Llama-3.2-1B | 0.024 | 0.026 | 0.376 | 0.926 | 2.460 | 8.160 |
| **Configuration: (512, 128)** | | | | | | |
| LLama-3.2-1B-ARMT | 0.147 | 0.574 | 1.15 | 2.29 | 4.52 | 8.98 |
| Diagonal Batching: LLama-3.2-1B-ARMT | 0.283 x0.52 | 0.248 x2.32 | 0.454 x2.53 | 0.861 x2.66 | 1.67 x2.71 | 3.3 x2.72 |
| **Configuration: (1024, 128)** | | | | | | |
| LLama-3.2-1B-ARMT | 0.149 | 0.291 | 0.578 | 1.15 | 2.3 | 4.48 |
| Diagonal Batching: LLama-3.2-1B-ARMT | 0.119 x1.25 | 0.196 x1.49 | 0.351 x1.65 | 0.656 x1.75 | 1.27 x1.81 | 2.48 x1.81 |
| **Configuration: (2048, 128)** | | | | | | |
| LLama-3.2-1B-ARMT | 0.094 | 0.177 | 0.344 | 0.679 | 1.35 | 2.68 |
| Diagonal Batching: LLama-3.2-1B-ARMT | 0.108 x0.87 | 0.176 x1.01 | 0.304 x1.13 | 0.571 x1.19 | 1.11 x1.22 | 2.18 x1.23 |
| **Configuration: (4096, 128)** | | | | | | |
| LLama-3.2-1B-ARMT | 0.082 | 0.155 | 0.301 | 0.594 | 1.18 | 2.35 |
| Diagonal Batching: LLama-3.2-1B-ARMT | 0.102 x0.80 | 0.172 x0.90 | 0.295 x1.02 | 0.553 x1.07 | 1.07 x1.10 | 2.1 x1.12 |

Table 2: Diagonal batching speeds up the execution - from 1.1 to 1.3 times comparing to base ARMT for 131072 sequence length, LLama-3.2-3B-ARMT, measured on Nvidia A100 GPU.

| Method | Sequence Length | | | | | |
|---|---|---|---|---|---|---|
| | 4096 | 8192 | 16384 | 32768 | 65536 | 131072 |
| Llama-3.2-3B | 0.168 | 0.344 | 0.769 | 1.95 | 5.59 | 18.2 |
| **Configuration: (1024, 128)** | | | | | | |
| LLama-3.2-3B-ARMT | 0.272 | 0.537 | 1.05 | 2.02 | 4.09 | 8.23 |
| Diagonal Batching: LLama-3.1-3B-ARMT | 0.274 x0.99 | 0.454 x1.18 | 0.833 x1.26 | 1.58 x1.28 | 3.1 x1.32 | 6.14 x1.34 |
| **Configuration: (4096, 128)** | | | | | | |
| LLama-3.2-3B-ARMT | 0.203 | 0.39 | 0.765 | 1.52 | 3.01 | 6.01 |
| Diagonal Batching: LLama-3.2-3B-ARMT | 0.239 x0.85 | 0.411 x0.95 | 0.739 x1.04 | 1.4 x1.09 | 2.72 x1.11 | 5.37 x1.12 |

implementations for all architectures. Flash Attention 2 for non-linear transformers, ARMT and ARMT with Diagonal Batching. We used mamba-ssm package for Mamba, and flash linear attention for RWKV.

## 4.4 DIAGONAL BATCHING VS MINI-BATCHING

Another way to increase compute load on GPUs is to increase the batch size. We evaluate the effectiveness of Diagonal Batching compared to standard mini-batching by measuring compute time per segment under identical hardware and model configurations. As shown in Figure 5, diagonal batching achieves compute scaling per segment that closely matches micro-batching across almost all tested scenarios.

To provide an upper bound on achievable performance, we also report the Ideal Even Load case, where all segments are computed with a full grouped layer with maximum achievable FLOPS. One can see that this even load setup is much better, mostly matching or overcoming the biggest batch sizes. The gap between them is our current implementation inefficiency. Notably, Diagonal Batching delivers substantial performance improvements for larger models (starting from 1B parameters), particularly when segment sizes are moderate. For these configurations, Diagonal Batching matches large batch sizes.

These findings suggest that Diagonal Batching effectively captures the utilization benefits of large-batch inference — through parallelized scheduling rather than increased memory allocation.

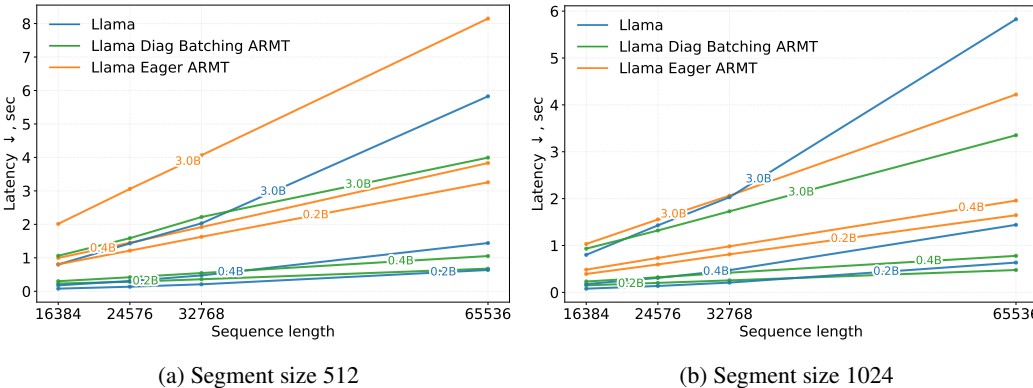

(a) Segment size 512

(b) Segment size 1024

Figure 3: Diagonal batching lowers the scaling curve for the whole family of LLaMA models. For a 512 segment size, the 3B model with Diagonal Batching is performing almost as 0.4B with Eager implementation. Model's family scaling for non-linear transformer, ARMT before and after Diagonal Batching usage. Measurements are done with bfloat16 on a single Nvidia A100 80Gb.

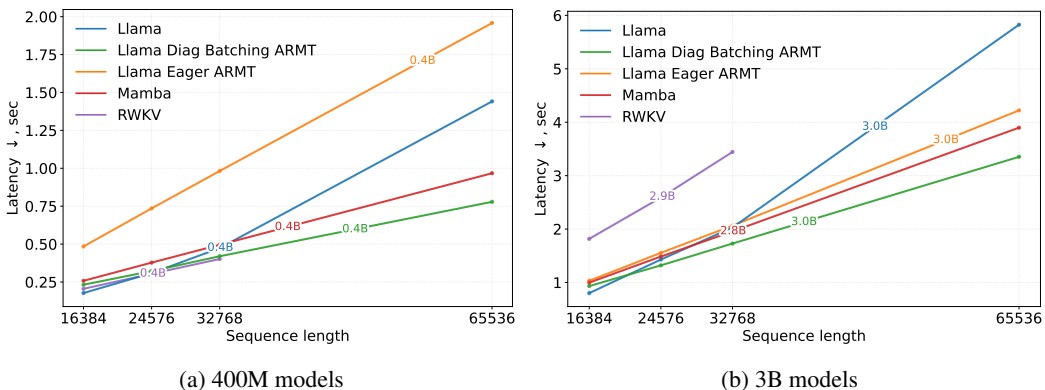

(a) 400M models

(b) 3B models

Figure 4: Diagonal batching makes ARMT the best performing linear architecture across different model sizes. Comparison between linear architectures and quadratic transformer grouped by the model's sizes. Measured with bfloat16 on Nvidia A100.

### 4.5 ERROR ACCUMULATION

We conducted an empirical investigation on computational error accumulation during the inference stage with Diagonal Batching. Our experiments show that the overall error is less than 2% for all sequences shorter than 32,768 tokens. This is comparable to other efficient layer implementations used in production. For example, we observed FlashAttention2 (Dao, 2024) gives 1-2% relative logits error compared to other attention implementations on the same input sequences.

The detailed error values for each segment are presented in Table 3. The error is calculated as the ratio of the Frobenius norm of the difference between the logits of the base ARMT implementation and the logits of ARMT with Diagonal Batching to the norm of the logits of the base ARMT. However, we find that effect of error accumulation on downstream tasks is negligible. To prove this, we evaluated the trained ARMT model both in original implementation and with Diagonal Batching; the results are presented in Table 4 in Appendix D. These results show that both implementations achieve the same results on the BABILong benchmark (Kuratov et al., 2024), while Table 5 in Appendix D shows that Diagonal Batching can increase the relative speed by up to 3.2x for 64k-length token sequences.

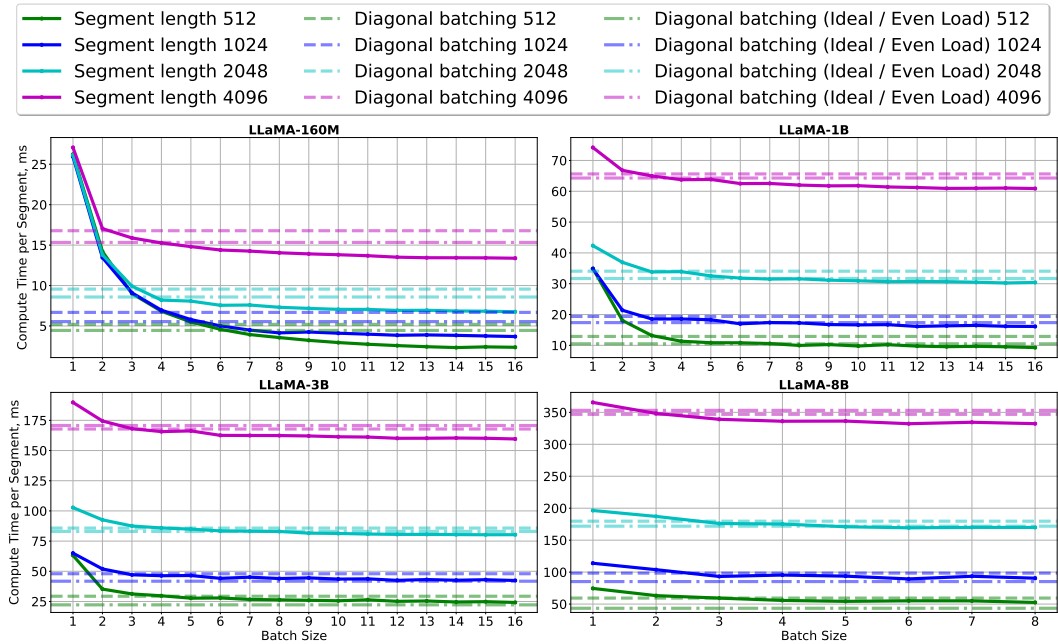

Figure 5: Ideal batch-size scaling vs grouped batching on Nvidia A100 for LLaMA models, time per segment in batch (group).

Table 3: During inference with diagonal batching, error accumulates but does not exceed 2%, which is comparable to the change of attention implementation (FlashAttention vs SDPA). The results for ARMT with Llama-3.2-1B-Instruct are shown with a segment size of 1024 tokens.

| Number of segments | 1 | 2 | 4 | 8 | 16 | 32 | 64 | 128 |
|---|---|---|---|---|---|---|---|---|
| Diagonal Batching, Error, % | 0.00 | 1.10 | 1.16 | 1.22 | 1.26 | 1.27 | 1.29 | 1.37 |
| FlashAttention2 (Dao, 2024) vs torch SDPA, Error, % | 1.25 | 1.15 | 1.17 | 1.22 | 1.36 | 1.45 | 1.79 | 2.04 |

## 5 CONCLUSIONS

We showed that the principal bottleneck in RMTs and their layer-memory variants (PRMTs) is not algorithmic complexity but *scheduling*: recurrent dependencies force fine-grained synchronization, which underutilizes modern accelerators. We introduced *Diagonal Batching*, a simple but powerful scheduling scheme that reorganizes the layer-segment computation grid into concurrency-friendly diagonals, enabling up to $N\_layers$ operations per kernel without altering exact recurrence. For single-request long-context inference (batch=1) on A100/H100, Diagonal Batching narrows the utilization gap without custom kernels, reducing cost per million tokens.

Relative to other linear-recurrent models, a base ARMT implementation is latency-inefficient. With Diagonal Batching, however, ARMT shows the *best latency scaling with context length*: latency grows near-linearly with length and matches or exceeds the end-to-end latency of custom-kernel baselines such as Mamba and RWKV at longer contexts. Compared to full-attention models, on LLaMA-1B at 131,072 tokens, ARMT with Diagonal Batching achieves $3.3\times$ lower latency than full-attention LLaMA-1B and $1.8\times$ over a sequential ARMT baseline, while preserving numerical fidelity on the same level as FlashAttention (about 1% relative logit error).

Considering these advantages, with right scheduling, Diagonal Batching turns theoretically appealing compute scaling of PRMTs into a practical solution for exact linear-time inference on extremely long contexts. By eliminating the major performance barrier, it positions memory-augmented recurrent Transformers as a competitive and scalable foundation for next-generation LLM applications that require efficient long-range input processing.

## REPRODUCIBILITY STATEMENT

To ensure reproducibility of results, we are releasing the full codebase. Currently, the code can be found in the Supplementary Materials. Section C provides details on reproducibility, including used hardware, software, and models details.

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

## A  THE USE OF LARGE LANGUAGE MODELS (LLMs)

LLMs were used exclusively for text polishing and editing (wording, spell checking).

## B  LIMITATIONS

Despite its advantages, Diagonal Batching has several limitations. First, it is not directly compatible with the Recurrent Memory Transformers (RMTs) due to intra-layer recurrence. However, a more promising approach is to focus on Parallel RMTs, which have already been shown in previous works to be more effective (Rodkin et al., 2024). Second, our current implementation assumes a uniform layer configuration. When models employ heterogeneous layers or varied hidden sizes, applying the technique requires more intricate grouping logic and manual engineering. Finally, the achievable speedup increases with the number of layers. Therefore, shallower models or models with very few layers will only see modest performance gains.

## C  REPRODUCIBILITY

We attach an anonymized repository containing all inference/training code, experiment scripts, and figure notebooks. Experiments were run on a single NVIDIA A100 80 GB (and verified on H100) with PyTorch 2.5.1, CUDA 12.1, and BF16 precision. Exact package versions are pinned in requirements.txt in the artifact. Code can be found in Supplementary Materials.

We evaluate LLaMA-3 and GPT ARMT variants with parameter sizes from approximate groups in 200M, 400M, 1-2B, 3B. These groups follow publicly available checkpoints for GPT, LLaMA, and linear transformers (Mamba and RWKV). All ARMT checkpoints and conversion utilities follow the baseline repository instructions, which our artifact pins (commit hash is included in the README). Unless stated otherwise, we use the following parameters for experiments: single request (batch=1), segment sizes in 512, 1024, 2048, 4096, memory tokens = 16 for the main latency results and bfloat16. BABILong experiments use the task configs described by the benchmark authors.

To reproduce results for papers, see the attached repository:

1. 'paper_experiments/measure_flops.ipynb' - individual operation scaling

2. 'paper_experiments/llamas_batch_scaling.ipynb' - LLaMA scaling with batch size

3. 'paper_experiments/ideal_grouped_scaling.ipynb' - reproduce Ideal/Even Load baseline in paper

4. 'usage_llama1b.ipynb' - performance comparison of torch model, ARMT implementation, and grouped batching algorithm

To reproduce the BABILong evaluation and training

1. Install additional dependencies - clone BABILong repo and prepare data:
   (a) 'git clone https://github.com/booydar/babilong.git'
   (b) 'unzip ./babilong/data/tasks_1-20_v1-2.zip'

2. 'run_eval_bl_fast_trained.py' - example of evaluation on BABILong for ARMT and ARMT with Diagonal batching (trained on BABILong train set)

3. 'calc_babilong_scores.ipynb' - extract and plot tables with BABILong scores and inference time for ARMT and ARMT with Diagonal batching

4. 'train_babilong_example.ipynb' - example of finetuning ARMT with Diagonal batching on BABILong

5. 'run_eval_bl_fast_finetuned.py' - example of evaluation on BABILong for ARMT with Diagonal batching after additional finetuning

Diagonal Batching benefits from base model kernel work optimizations, which are assumed to be presented (we use torch models from Huggingface Transformers); no custom CUDA is required.

Table 4: Diagonal Batching maintains the same scores as the original ARMT inference method on the BABILong benchmark. Scores of the models were evaluated on the first two tasks: QA1 and QA2.

| Task | Length, tokens | LLama-3.2-1B ARMT | LLama-3.2-1B ARMT, Diagonal Batching |
|------|------|------|------|
| QA1 | 0K | 100 | 100 |
| | 1K | 100 | 100 |
| | 2K | 100 | 100 |
| | 4K | 100 | 100 |
| | 8K | 100 | 100 |
| | 16K | 100 | 100 |
| | 32K | 100 | 100 |
| | 64K | 70 | 69 |
| | 128K | 4 | 4 |
| QA2 | 0K | 100 | 100 |
| | 1K | 100 | 100 |
| | 2K | 100 | 100 |
| | 4K | 100 | 100 |
| | 8K | 99 | 100 |
| | 16K | 98 | 98 |
| | 32K | 94 | 94 |
| | 64K | 47 | 46 |
| | 128K | 3 | 3 |

From grouped GEMM, we use CUTLASS GroupedGemm to avoid separate concatenation of input segments (it is done implicitly by allocating the output blob as continuous memory).

We fix PyTorch/CUDA seeds and enable deterministic flags where possible; minor variance in end-to-end latency is expected due to kernel autotuning and GPU clocks. Seeds and flags are set in the provided scripts.

As a result, in the attachment are provided artifacts, including source code, pinned requirements, ARMT patch + commit, run scripts, notebooks producing all plots/tables, and guidance commands used for each result.

## D  EVALUATING MODELS WITH DIAGONAL BATCHING

Although diagonal Batching significantly speeds up the inference, it also introduces some numerical drifts due to the optimized execution procedure. To estimate the effect of these drifts on practical tasks, we evaluated the ARMT model on the BABILong benchmark Kuratov et al. (2024) with and without diagonal Batching. The ARMT model was trained on the BABILong dataset with curriculum learning on length up to 8192 tokens, similar to the approach described in Kuratov et al. (2024). After, we evaluated this model with and without diagonal batching on QA1 and QA2 tasks from BABILong. Note that we did not change the weights of the model in this experiment; we simply applied the proposed Diagonal Batching grouping method.

The evaluation results are presented in Table 4. As one can see, despite the numerical drifts during the forward pass, the generation results remain almost unchanged up to the 65536 input length. These results show that diagonal batching preserves the quality of the generation of the trained ARMT model and can be used as a drop-in replacement to speed up the inference.

We also compared the inference time of these two approaches on the same benchmark. In this experiment, we measure not the forward pass time, but the generation time on the BABILong. Table 5 shows that the diagonal batching approach significantly speeds up the generation, up to 3 times on the input length of 65536 tokens. During both of these experiments, we used the following ARMT configuration - the size of the segment was set to 1024 tokens, the number of memory tokens was set to 16, and the associative memory hidden size was 64.

Finally, we implemented the backward pass for diagonal batching to support training. Aligning the training and inference code eliminates a discrepancy that is likely the source of logit-level floating-point drift.

Table 5: Diagonal Batching significantly speeds up ARMT inference on longer inputs. Inference time (in seconds) and relative speed-up of the models are given on the BABILong dataset, first two tasks.

| Task | Length, tokens | LLama-3.2-1B, ARMT | LLama-3.2-1B, ARMT, Diagonal Batching | Speed-up ($\times$ times) |
|---|---|---|---|---|
| QA1 | 2K | 13.43 | 15.06 | 0.89 |
| | 4K | 22.45 | 17.99 | 1.25 |
| | 8K | 41.41 | 22.49 | 1.84 |
| | 16K | 79.16 | 33.12 | 2.39 |
| | 32K | 153.68 | 54.20 | 2.84 |
| | 64K | 302.15 | 94.36 | 3.20 |
| QA2 | 2K | 13.08 | 14.93 | 0.88 |
| | 4K | 22.66 | 18.21 | 1.24 |
| | 8K | 41.66 | 22.70 | 1.84 |
| | 16K | 79.80 | 33.38 | 2.39 |
| | 32K | 153.82 | 53.46 | 2.88 |
| | 64K | 303.40 | 94.69 | 3.20 |

To further evaluate the difference between ARMT model with and without Diagonal Batching, we calculated how many tokens differ among tokens chosen by argmax during forward pass. The results are presented in Table 6.

Table 6: During inference with diagonal batching, error accumulates in chosen by argmax tokens, but does not exceed 2%. The results for ARMT with Llama-3.2-1B-Instruct are shown with a segment size of 1024 tokens.

| Number of segments | 1 | 2 | 4 | 8 | 16 | 32 | 64 | 128 |
|---|---|---|---|---|---|---|---|---|
| Diagonal Batching, percentage of different tokens chosen by argmax | 0.00 | 0.05 | 0.02 | 0.05 | 0.09 | 0.12 | 0.12 | 0.13 |

## D.1 LINEAR LAYER EFFICIENCY

The only change from the base model is that we substitute linear layer with matrix multiplication to layers with grouped GEMM with the group equal to all weights of the linear layers. In Figure 6 we show that grouped GEMM FLOPS scales similarly through group size to GEMM with the corresponding batch size. This gives the basis that our method should scale similarly to the underlying model with batch size, as all other operations are basically the same (but in a different order).

Second, we have a group size equal to the number of layers in the model. This way, we move the grouped GEMM operation to peak GEMM flops for a100 and h100 GPUs, ensuring high utilization. Corresponding FLOPS improvement shown in Figure 6.

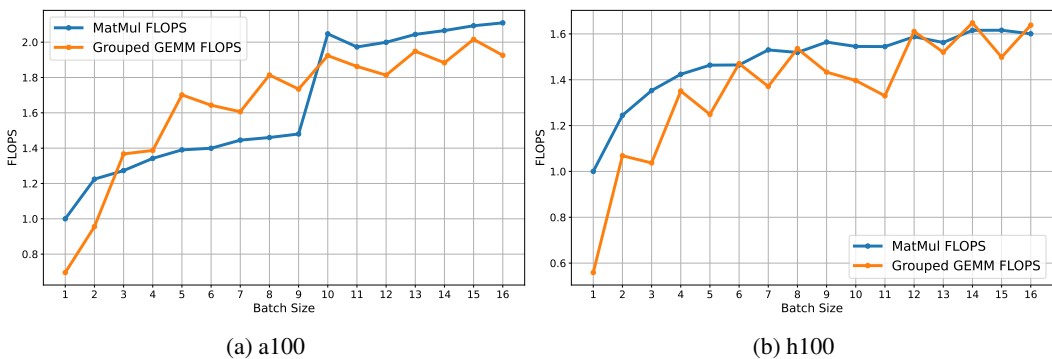

(a) a100              (b) h100

Figure 6: Cutlass Group GEMM scales similarly to batch size 1 Linear layer's matrix multiplication, starting from group size 4.

## D.2 ATTENTION LAYER EFFICIENCY

Our method does not modify the attention layer at all. Instead, attention just performs a batched operation with a batch size equal to the number of layers. This increases its performance to the implementation FLOPS peak. We show relative FLOPS speedups in Figure 7.

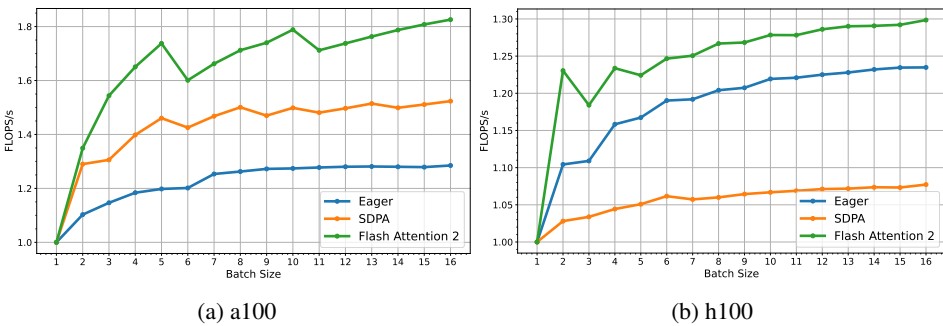

(a) a100                               (b) h100

Figure 7: Diagonal batching increases attention performance by treating groups as batches—similar to increasing the model's overall batch size.

# E ADDITIONAL MEASUREMENTS

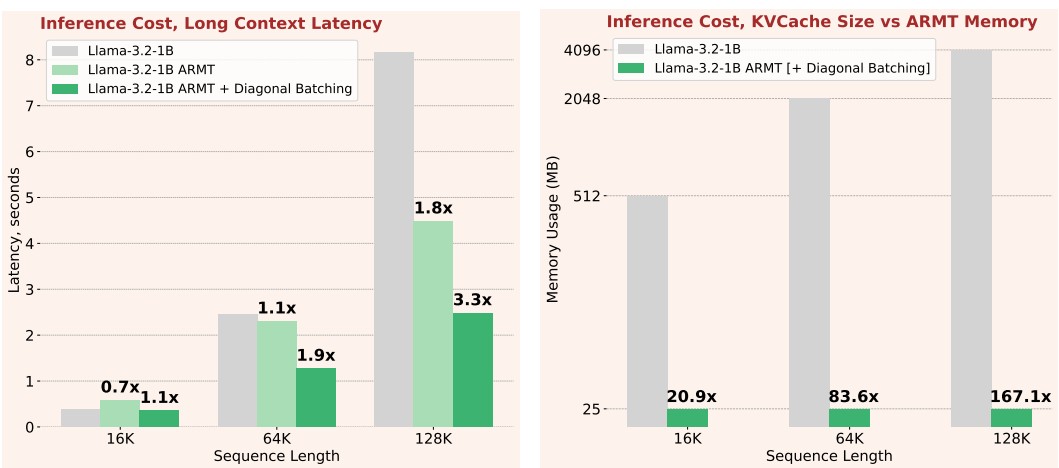

Figure 8: **Diagonal Batching enables the Recurrent Memory Transformers (ARMT) to process 128k token sequences 3.3x faster than the LLama-3.2-1B model, with 167.1x memory savings.** These results were obtained using an A100 GPU, and the segment size for the ARMT was set to 1,024 tokens.

To clearly illustrate the speedup provided by the developed diagonal batching algorithm, we present relative improvements across various configurations and sequence lengths. Results for speedup against the original ARMT implementation are shown in Table 11 and against the underlying LLaMA model in Table 10. These measurements provide additional insights into how our method scales and compares to the baseline implementations.

We also present results for different size models of LLaMA-3 family Grattafiori et al. (2024): LLaMA-160M (Table 9), 1B (Table 1), 3B (Table 7), and 8B (Table 8) models.

Table 7: Diagonal batching speeds up the execution - from 1.1 to 1.3 times comparing to base ARMT for 131072 sequence length, LLama-3.2-3B-ARMT, measured on Nvidia A100 GPU.

| Method | Sequence Length | | | | | |
|---|---|---|---|---|---|---|
| | 4096 | 8192 | 16384 | 32768 | 65536 | 131072 |
| Llama-3.2-3B | 0.168 | 0.344 | 0.769 | 1.95 | 5.59 | 18.2 |
| **Configuration: (1024, 128)** | | | | | | |
| LLama-3.2-3B-ARMT | 0.272 | 0.537 | 1.05 | 2.02 | 4.09 | 8.23 |
| Diagonal Batching: LLama-3.1-3B-ARMT | 0.274 x0.99 | 0.454 x1.18 | 0.833 x1.26 | 1.58 x1.28 | 3.1 x1.32 | 6.14 x1.34 |
| **Configuration: (4096, 128)** | | | | | | |
| LLama-3.2-3B-ARMT | 0.203 | 0.39 | 0.765 | 1.52 | 3.01 | 6.01 |
| Diagonal Batching: LLama-3.2-3B-ARMT | 0.239 x0.85 | 0.411 x0.95 | 0.739 x1.04 | 1.4 x1.09 | 2.72 x1.11 | 5.37 x1.12 |

Table 8: Diagonal batching speed-ups the execution - from 1.05 to 1.14 times comparing to base ARMT for 131072 sequence length. Execution time comparison (in seconds) and relative speedups across different sequence lengths compared to LLama-3.2-8B-ARMT. Configuration in format (segment_size, memory_tokens). Nvidia A100 GPU.

| Method | Sequence Length | | | | | |
|---|---|---|---|---|---|---|
| | 4096 | 8192 | 16384 | 32768 | 65536 | 131072 |
| Llama-3.1-8B | 0.332 | 0.682 | 1.48 | 3.61 | 9.82 | 30.4 |
| **Configuration: (1024, 128)** | | | | | | |
| LLama-3.1-8B-ARMT | 0.497 | 0.936 | 1.82 | 3.63 | 7.22 | 14.4 |
| Diagonal Batching: LLama-3.1-8B-ARMT | 0.478 x1.04 | 0.86 x1.09 | 1.64 x1.11 | 3.2 x1.13 | 6.34 x1.14 | 12.6 x1.14 |
| **Configuration: (4096, 128)** | | | | | | |
| LLama-3.1-8B-ARMT | 0.384 | 0.754 | 1.48 | 2.95 | 5.86 | 11.7 |
| Diagonal Batching: LLama-3.1-8B-ARMT | 0.432 x0.89 | 0.781 x0.97 | 1.46 x1.01 | 2.83 x1.04 | 5.6 x1.05 | 11.1 x1.05 |

Table 10: Diagonal batching ARMT implementation allows to speedup the execution for longer sequences due to linear complexity - from 2.4 times to 3.8 times with respect to LLama-3.2-1B for 131072 sequence length. Table shows Diagonal Batching executor speedup against original LLama-3.2-1B for different methods across sequence lengths. Configuration in format (segment_size, memory_tokens). Measured on Nvidia A100 GPU.

| Method | Sequence Length | | | | | |
|---|---|---|---|---|---|---|
| | 4096 | 8192 | 16384 | 32768 | 65536 | 131072 |
| LLama-3.2-1B, configuration: (512, 128) | 0.085 | 0.105 | 0.828 | 1.075 | 1.473 | 2.473 |
| LLama-3.2-1B, configuration: (1024, 128) | 0.202 | 0.133 | 1.071 | 1.412 | 1.937 | 3.290 |
| LLama-3.2-1B, configuration: (2048, 128) | 0.222 | 0.148 | 1.237 | 1.622 | 2.216 | 3.743 |
| LLama-3.2-1B, configuration: (4096, 128) | 0.235 | 0.151 | 1.275 | 1.675 | 2.299 | 3.886 |

Table 9: Diagonal batching speed-ups the execution - from 1.6 to 3.9 times comparing to base ARMT for 131072 sequence length. Execution time comparison (in seconds) and relative speedups across different sequence lengths compared to LLama-160M-ARMT. Configuration in format (segment_size, memory_tokens). Nvidia A100 GPU.

| Method | Sequence Length | | | | | |
|---|---|---|---|---|---|---|
| | **4096** | **8192** | **16384** | **32768** | **65536** | **131072** |
| Llama-160M | 0.017 | 0.033 | 0.075 | 0.196 | 0.594 | 2.03 |
| **Configuration: (1024, 128)** | | | | | | |
| LLama-160M-ARMT | 0.105 | 0.211 | 0.422 | 0.877 | 1.72 | 3.37 |
| Diagonal Batching: LLama-160M-ARMT | 0.061 x1.72 | 0.087 x2.43 | 0.138 x3.06 | 0.243 x3.61 | 0.451 x3.81 | 0.855 x3.94 |
| **Configuration: (4096, 128)** | | | | | | |
| LLama-160M-ARMT | 0.031 | 0.057 | 0.111 | 0.216 | 0.432 | 0.855 |
| Diagonal Batching: LLama-160M-ARMT | 0.046 x0.67 | 0.062 x0.92 | 0.094 x1.18 | 0.156 x1.38 | 0.284 x1.52 | 0.537 x1.59 |

Table 11: Diagonal batching allows to speedup the execution for longer sequences - from 1.1 times to 2.7 times with respect to base ARMT for 131072 sequence length. In cases when diagonal batching is slower, we can fall back to the original inference algorithm at runtime. Table shows Diagonal Batching executor speedup against original ARMT inplementation for different methods across sequence lengths. Configuration in format (segment_size, memory_tokens). Measured on Nvidia A100 GPU.

| Method | Sequence Length | | | | | |
|---|---|---|---|---|---|---|
| | **4096** | **8192** | **16384** | **32768** | **65536** | **131072** |
| LLama-3.2-1B, configuration: (512, 128) | 0.519 | 2.315 | 2.533 | 2.660 | 2.707 | 2.721 |
| LLama-3.2-1B, configuration: (1024, 128) | 1.252 | 1.485 | 1.647 | 1.753 | 1.811 | 1.806 |
| LLama-3.2-1B, configuration: (2048, 128) | 0.870 | 1.006 | 1.132 | 1.189 | 1.216 | 1.229 |
| LLama-3.2-1B, configuration: (4096, 128) | 0.804 | 0.901 | 1.020 | 1.074 | 1.103 | 1.119 |

## E.1 APPLICATION TO OTHER MODELS

Diagonal Batching may also benefit other PRMT models as they contain recurrent structure. Examples include Mamba and xLSTM. In these cases, diagonal batching can be applied with a segment size of 1, since these models exhibit token-level recurrence.

In Table 12 and Figure 9, we show that diagonal batching provides higher efficiency per segment by increasing compute parallelism.

| | Batch size | | | | | | | |
|---|---|---|---|---|---|---|---|---|
| Method | 1 | 2 | 4 | 8 | 12 | 16 | 20 | 24 |
| mamba_ssm with DB | **0.0000382** | **0.000038** | **0.000038** | **0.000038** | **0.000038** | **0.000038** | **0.000038** | **0.000038** |
| mamba_ssm | 0.00092 | 0.00046 | 0.00023 | 0.000115 | 0.000092 | 0.000059 | 0.000046 | 0.000039 |
| No mamba_ssm with DB | **0.002865** | **0.002865** | **0.002865** | **0.002865** | **0.002865** | **0.002865** | **0.002865** | **0.002865** |
| No mamba_ssm | 0.0658 | 0.03547 | 0.01778 | 0.00929 | 0.005743 | 0.00424 | OOM | OOM |

Table 12: Diagonal batching can be beneficial for Mamba on small batch sizes. Mamba throughput/latency (in seconds) across batch sizes with and without efficient CUDA kernels for mamba ($mamba\_ssm$). Measured on state-spaces/mamba-130m-hf model on single A100 inference. 8k context used for measurements to prevent OOM on large batch sizes and, as mamba is token-recurrent model, its efficiency does not increase then context is larger.

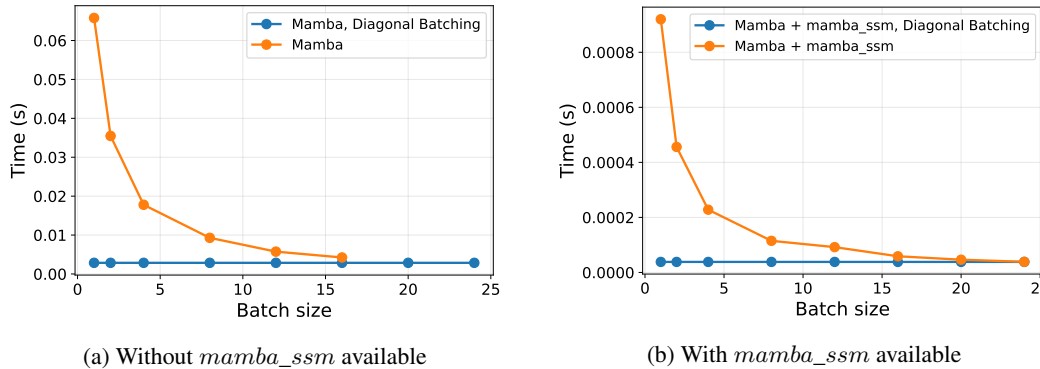

(a) Without $mamba\_ssm$ available

(b) With $mamba\_ssm$ available

Figure 9: Comparison of Mamba performance with and without efficient CUDA kernels for mamba ($mamba\_ssm$).

To apply diagonal batching to Mamba in practice, one must rewrite the Mamba CUDA kernels to enable external segmentation of the forward pass, as the current implementation computes the full forward for each layer.

### E.2 ADDITIONAL COMPARISON WITH OTHER ARCHITECTURES

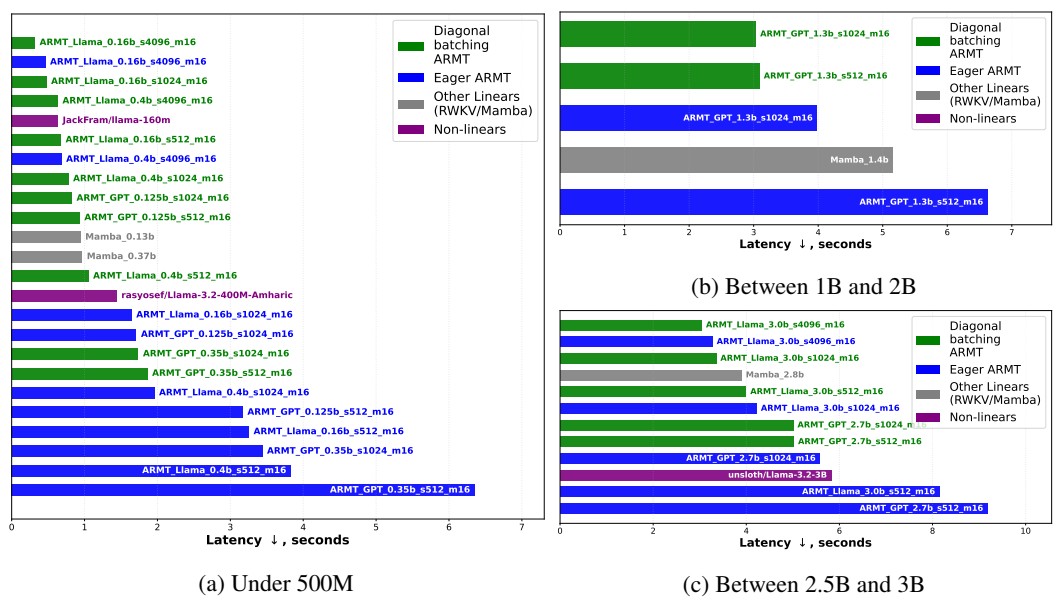

(a) Under 500M

(b) Between 1B and 2B

(c) Between 2.5B and 3B

Figure 10: ARMT with Diagonal Batching is the best latency model in each category for the 64k context. ARMT+Diagonal Batching has very competitive performance across a wide variety of segment sizes. Comparison is made for open source models that can out of the box support such context. Reference efficient implementation is used - mamba-ssm for Mamba and flash linear attention Yang & Zhang (2024) for RWKV. Single Nvidia A100 80Gb for measurements.

### E.3 DECODING STAGE WITH ARMT

Diagonal batching does not modify the decoding stage, meaning that ARMT inference with diagonal batching remains identical to the native ARMT implementation. However, ARMT provides several advantages over standard quadratic-time Transformers. Most importantly, it eliminates the need to store and repeatedly move large KV-caches between HBM and registers for each request - the cost that grows linearly with both load and context length. Instead, ARMT relies on compact associative

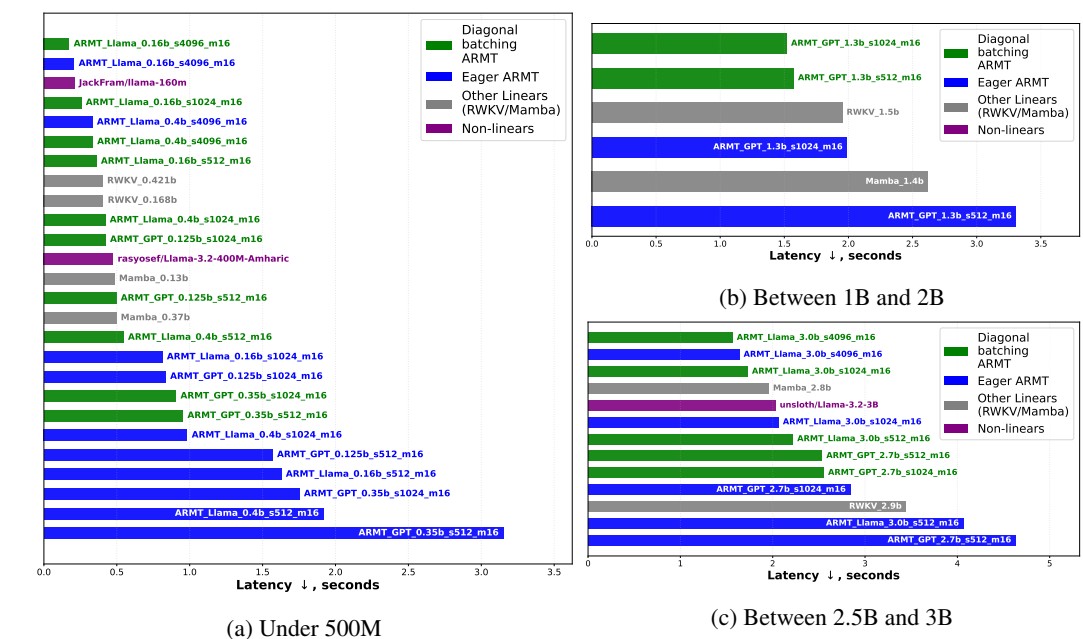

(a) Under 500M

(b) Between 1B and 2B

(c) Between 2.5B and 3B

Figure 11: ARMT with Diagonal Batching is the best latency model in each category for the 32k context. ARMT+Diagonal Batching has very competitive performance across a wide variety of segment sizes. Comparison is made for open source models that can out of the box support such context. Reference efficient implementation is used - mamba-ssm for Mamba and flash linear attention for RWKV. Single Nvidia A100 80Gb for measurements.

memory produced during the prefill stage. This memory has fixed size per context and therefore scales only with the number of concurrent requests. As a result, ARMT can execute substantially more decoding phases in parallel within a disaggregated prefill–decode inference pipeline.

Table 13 shows that LLaMA-1B with ARMT sustains far more parallel requests before reaching OOM on an NVIDIA RTX 6000, and even very long contexts remain feasible.

Table 14 demonstrates that ARMT maintains stable decoding efficiency across large batch sizes, reducing memory-boundedness during decode. Further optimization of memory kernels for ARMT remains an open direction.

| Context Length | Max Batch LLaMA | Max Batch LLaMA ARMT |
|---|---|---|
| 4096 | 4 | 16 |
| 8192 | 1 | 16 |
| 16384 | 0 | 16 |
| 32768 | 0 | 16 |
| 65536 | 0 | 16 |

Table 13: ARMT allows to handle constant big batch size in parallel on decode, not depending on input context size. Maximum batch size before OOM on NVIDIA RTX 6000.

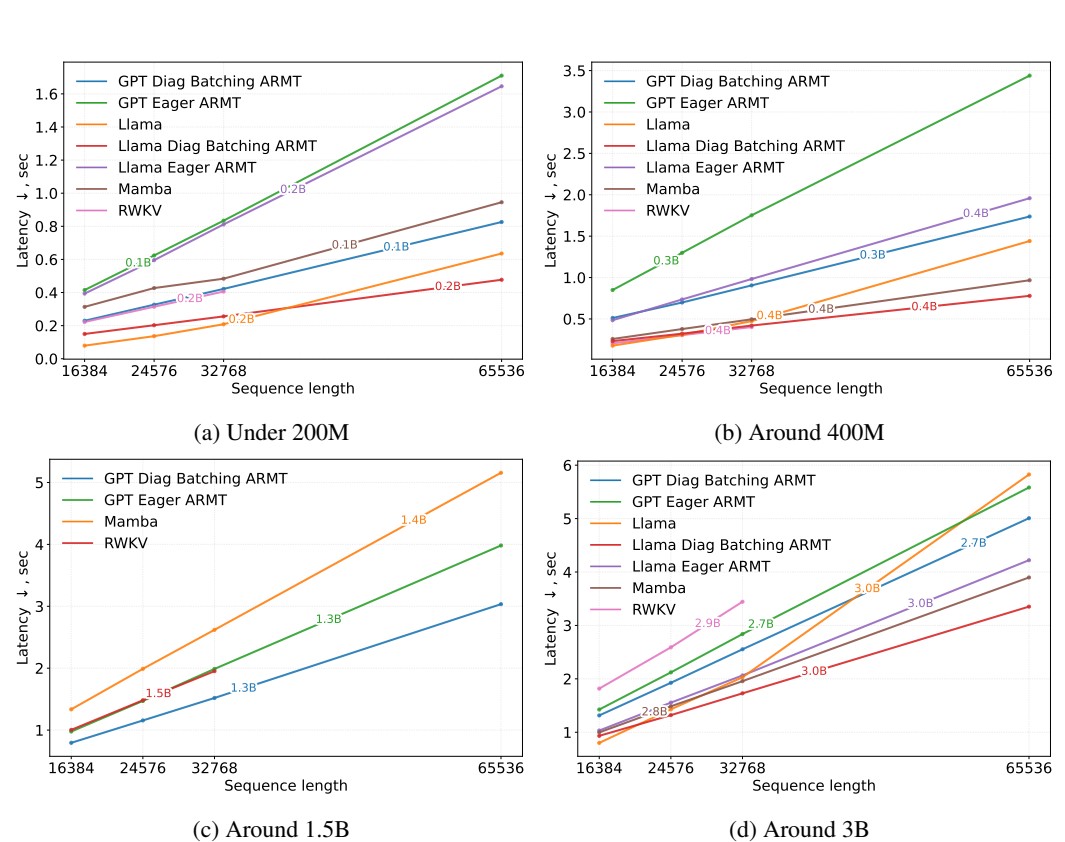

(a) Under 200M  (b) Around 400M

(c) Around 1.5B  (d) Around 3B

Figure 12: Plot comparison across different architectures grouped by model size.

| Prefill Size | Decode 10 Tokens LLaMA, s | Decode 10 Tokens ARMT LLaMA, s |
|---|---|---|
| 1024 | 0.007 | 0.025 |
| 4096 | 0.006 | 0.025 |
| 8192 | 0.007 | 0.081 |
| 16384 | OOM | 0.027 |
| 32768 | OOM | 0.026 |
| 65536 | OOM | 0.027 |

Table 14: Decode latency does not grow while batch size increase for ARMT model, yet it can handle much more requests in paralle then classic LLaMA transformer. Decode 10-token runtime comparison between LLaMA and ARMT-LLaMA for different prefill sizes. Measured on 48Gb NVIDIA RTX 6000.

