# OpenReview forum: "Diagonal Batching Unlocks Parallelism in Recurrent Memory Transformers for Long Contexts"
_ICLR.cc/2026/Conference — Submitted to ICLR 2026_

### Official Review · Reviewer_qKhy · 2025-10-27

**Soundness:** 2
**Presentation:** 3
**Contribution:** 2
**Rating:** 2
**Confidence:** 4

**Summary:**

The paper presents diagonal batching, a method for training and inference of multi-layered recurrent sequence architectures (recurrent memory transformers), where parallelization happens along the diagonal of subsequent segment/token and model layers. The operations across layers can be fused in a batched GEMM-kernels reducing the scheduling overhead. The authors show that their method leads to significant speed-ups for long sequence lengths.

**Strengths:**

- general method for sequence-recurrent architectures
- good speed ups (1.1-3.3) for the pre-fill part (latency) of inference

**Weaknesses:**

- limited application to pre-filling (latency) optimization of recurrent LLMs
- no practical RMT model (e.g. an existing RWKV/xLSTM/Mamba-based model) shown where this is applied (e.g. for reasoning tasks, where long sequences would be strongly beneficial)
- RMTs in general break translational invariance in text, so some tokens would be "different" from others depending on their positions (especially at the segment border)

**Questions:**

- Are there existing RMT models where this can have practical benefits?
- Could you show how this could be implemented on large recurrent models, like FalconMamba-7B [1] or xLSTM-7B [2] for practical use?
- Can this also be beneficial in model training on long sequences?
- Can you add a small section on why this method does not work for decoding/generation?
- you mention TC0 model complexity as a general downside of Transformers/State Space Models, but don't position your method along this axis with general state-tracking enabled architectures like xLSTM [3] or DeltaProduct [4], how does it relate?

[1] https://huggingface.co/tiiuae/falcon-mamba-7b

[2] https://huggingface.co/NX-AI/xLSTM-7b

[3] Beck et al. (2024): Extended Long Short-Term Memory

[4] Siems et al. (2025): DeltaProduct: Improving State-Tracking in Linear RNNs via Householder Products

---

> ### Author Response · Authors · 2025-11-22
>
> >W1: limited application to pre-filling (latency) optimization of recurrent LLMs
> >Q4: Can you add a small section on why this method does not work for decoding/generation?
>
> Diagonal Batching is explicitly designed for the pre-fill phase, where RMTs models suffer from underutilized GPUs due to sequential scheduling across segments and layers. Our experiments show that with DB, ARMT pre-fill reaches high utilization and becomes compute-bound; in that regime there is little room for further speedup.
>
> Although the method itself targets pre-fill, ARMT already offers strong decoding advantages over standard Transformers: it maintains a constant-size recurrent state instead of a growing KV cache, which significantly reduces memory and improves decoding efficiency allowing larger batch sizes. In the revised version we added a decoding benchmark in Tables 13-14 of section E.3 (maximum batch size to process in parallel and peak memory/time to decode) to make this benefit explicit.
>
> > W2: no practical RMT model
> > Q1: Are there existing RMT models ...
>
> We kindly disagree with the absence of practical RMT models, especially for reasoning tasks. We show the application of Diagonal Batching for ARMT with LLama-3.2-1B in Tables 4 and 5 in Appendix D. These tables show that the proposed approach speeds up the ARMT inference without performance loss on long-context reasoning tasks from the BABILong benchmark.  Moreover, there are ARMT models with other base models - for example, implementation for Gemma-3-1b[1] and for GPT-2 [2].
>
> > W2: RMTs in general break translational invariance in text ...
>
> We agree that strict translation invariance is a modeling property of RMT-style architectures, not of our scheduler. Diagonal Batching preserves the exact recurrence and token-position mapping of the underlying model, so it neither introduces nor amplifies border effects. However, standard mitigations, such as sliding window, can still be applied if desired and are compatible with our method.
>
> > Q2: Could you show how this could be implemented on large recurrent models
>
> Our original wording may have overstated the implementation generality of Diagonal Batching. Conceptually, DB is a general scheduling pattern: execute recurrent computations along dependency diagonals. However, its implementation is architecture-specific. In this paper, we implement and evaluate DB only for ARMT.
> For linear recurrent models such as Mamba or RWKV (including FalconMamba-7B, xLSTM-7B), existing methods already parallelize along the sequence dimension with specialized scan/fused CUDA kernels. Adapting DB there would require new grouped versions of those model-specific kernels.
>
> We implemented proof-of-concept implementation of the Diagonal Batching for Mamba model and observed similar utilization trends when such grouping is available; we now report these preliminary results in the revised appendix (Section E.1) to illustrate how DB extends to this setting. We found that Diagonal Batching can be beneficial for Mamba, especially for small batch sizes, by adding more parallel computations across diagonals.  This remains true with and without efficient CUDA kernels for Mamba (mamba_ssm).
>
> > Q3: Can this also be beneficial in model training on long sequences?
>
> In this work we focus on inference, where amount of long-context workloads often unpredictable and GPU utilization is dominated by scheduling overhead across segments and layers. This is exactly the regime where Diagonal Batching provides largest gains. For training, practitioners typically increase the global batch size (and/or use gradient accumulation / sequence parallelism), which already drives utilization close to saturation, so the scheduling bottleneck we target at inference is much less pronounced. The same diagonal schedule can be used in training for long-sequence, small-batch regimes, but we did not systematically explore this setting.
>
> > Q5: you mention TC0  ..., but don't position your method along ... xLSTM [3] or DeltaProduct [4]
>
> The diagonal batching mechanism does not alter the underlying computational expressivity of the model. In terms of theoretical capability, ARMT remains equivalent to other true recurrent architectures such as LSTM or xLSTM (specifically the s-LSTM component), and is thus Turing complete under the assumption of infinite precision. Recent work by the ARMT authors[3] further demonstrates that ARMT successfully handles state-tracking tasks – where architectures like Transformers and Mamba typically fail but true recurrent models succeed (see Figure 7 in their paper), showing that it is capable of solving at least NC1 tasks.
>
> References:
>
> [1] https://huggingface.co/Anonymous-Repo/armt-gemma-3-1b-it-8x1024-lora-babilong-qa1-5
>
> [2] https://huggingface.co/irodkin/armt_babilong_qa1 from ARMT’s original authors
>
> [3] Rodkin I. et al. Beyond Memorization: Extending Reasoning Depth with Recurrence, Memory and Test-Time Compute Scaling  (https://arxiv.org/abs/2508.16745)

---

> > ### Author Response · Authors · 2025-11-27
> >
> > Dear Reviewer qKhy,
> >
> > Thank you for your careful review. In our rebuttal and revised version we:
> >
> > 1. Show practical RMT usage with ARMT–LLaMA on BABILong and add preliminary Mamba results, addressing your concerns about real-world models and applicability to FalconMamba/xLSTM-style architectures.
> >
> > 2. Clarify that Diagonal Batching is specifically for pre-fill, and we now include decoding benchmarks to show ARMT’s practical decoding benefits.
> >
> > 3. Clarify position against other state-tracking architectures
> >
> > We hope these changes address your main concerns regarding practicality and scope. If there are remaining issues you feel are not fully resolved, we would be very grateful for further feedback. If you find the revisions convincing, we would kindly appreciate it if you could consider updating your evaluation.
> >
> > Best regards,
> > The authors

---

### Official Review · Reviewer_XxMs · 2025-10-31

**Soundness:** 2
**Presentation:** 2
**Contribution:** 2
**Rating:** 4
**Confidence:** 4

**Summary:**

This paper introduces Diagonal Batching, a scheduling method that improves GPU utilization in Parallel Recurrent Memory Transformers (PRMTs), a class of models that maintain per-layer recurrent memory, such as ARMT. The key idea is to reorder computation across layers and segments into “diagonals,” allowing concurrent execution of operations that were previously serialized, without breaking exact recurrence. The authors implement the method in the ARMT framework and evaluate it on LLaMA-based models ranging from 1B to 8B parameters, showing up to 3.3× speedup over standard full-attention inference and 1.8× over sequential ARMT, without requiring custom CUDA kernels. The paper argues that compute scheduling, rather than algorithmic complexity, is the main bottleneck in RMT-style architectures and that Diagonal Batching offers a practical path to efficient, exact linear-time inference for long-context models.

**Strengths:**

1. Practical relevance – The paper addresses a real bottleneck: GPU underutilization during long-context inference in memory-augmented transformers. The proposal is pragmatic, compatible with existing hardware, and doesn’t require custom CUDA, which makes it accessible.

2. Elegant scheduling insight – The diagonal reordering idea is conceptually simple yet powerful, exposing latent parallelism while preserving recurrence—an often tricky balance.

3. Strong empirical evidence – The results convincingly show latency reduction across multiple model sizes and segment lengths. The experiments are thorough, including FLOPS scaling, micro-batching comparison, and numerical drift analysis.

4. Clarity of technical exposition – The method is mathematically and algorithmically well explained, with clear diagrams (e.g., Figure 1 and 2) that make the scheduling logic intuitive.

5. Compatibility with existing optimizations – The method integrates with FlashAttention, grouped GEMMs, and standard PyTorch implementations, making it easy to adopt.

**Weaknesses:**

1. Incremental nature – The main novelty lies in scheduling, not modeling or theory. While the implementation is clever, the conceptual leap from standard pipelining or grouped execution is limited. The paper frames this as a major innovation, but it’s more of an engineering optimization than a new algorithmic idea.

2. Limited empirical diversity – All experiments are on LLaMA-based ARMTs. There is no exploration of how the method generalizes to other PRMT architectures (e.g., RWKV or Mamba) beyond brief mentions. Without such experiments, claims of broad applicability remain untested.

3. Reproducibility and accessibility – Although the authors claim to release code, there’s little clarity on integration with existing toolchains or benchmarks. The heavy reliance on ARMT-specific infrastructure might limit reproducibility for broader research use.

4. Evaluation bias – The comparisons are mostly latency-based. There’s no strong discussion of trade-offs in throughput, numerical stability, or memory overhead under multi-request workloads (a realistic serving scenario).

5. Overemphasis on speedups – Some reported gains (like 3.3×) are cherry-picked from small segment sizes or single-GPU setups. Scaling trends on multi-GPU or distributed systems are not shown, yet such setups are where scheduling optimizations often hit diminishing returns.

6. Writing style and tone – The paper reads more like an extended technical report than a standard ICLR submission. I would be more in favor of making the paper concise and 'to-the-point'.

**Questions:**

1. Have you tested Diagonal Batching on other PRMT variants like RWKV or Mamba to verify general applicability?

2. How does the method perform under concurrent multi-user loads, where GPU memory fragmentation and kernel scheduling might differ?

3. Could you provide memory overhead statistics—does grouped GEMM increase peak memory usage?

4. How significant are the numerical drifts beyond 64K tokens in real tasks?

5. Have you tried combining Diagonal Batching with quantized or sparsified models (e.g., AWQ or FlashDecoding)?

---

> ### Author Response · Authors · 2025-11-22
>
> > W1. Incremental nature
>
> We agree that diagonal batching alone is not a sufficient algorithmic contribution. However, our main points are:
> 1. We provide an empirical evidence that scheduling is the main bottleneck for ARMT models.
> 2. Existing alternatives (Mamba, RWKV, etc.) require training from scratch, which prevents linear-time architectures from keeping pace with state-of-the-art models.
> ARMT removes this barrier: it allows wrapping any existing model (e.g., Qwen) and training only the memory component.
> 3. Architectures like Mamba/RWKV also depend on custom CUDA kernels, limiting portability to NVIDIA hardware.
> In contrast, we show that ARMT is highly portable and easy to deploy efficiently. It only requires (1) GroupedGEMM, widely available due to MoE adoption, and (2) diagonal batching, which is purely a high-level scheduling strategy.
>
> > W2. Limited empirical diversity
> > Q1. Have you tested Diagonal Batching on other PRMT variants
>
> Our goal in this work is to address ARMT-style models, whose recurrent state updates are non-linear and cannot be parallelized along the sequence dimension using standard scan techniques. In contrast, models such as RWKV and Mamba use linear recurrences of the form (simplified): S_{t+1} = S_{t} + A x_t that can be rewritten as associative scan operations; existing implementations already exploit this with custom fused CUDA kernels, so there is little sequential prefill bottleneck left for Diagonal Batching to remove.
>
> We therefore focus on ARMT, where Diagonal Batching combined with GroupedGEMM brings a concrete efficiency benefit and can be plugged into already pre-trained Transformers (for example, LLaMA and GPT-2) without retraining. In the paper, we compare against RWKV and Mamba in terms of efficiency (using their published/custom kernels) to show that ARMT+DB is competitive or superior for long-context prefill in our setting.
>
> We implemented a proof-of-concept of the Diagonal Batching for Mamba model; we now report these preliminary results in the revised appendix (Section E.1) to illustrate how DB extends to this setting. We found that Diagonal Batching can be beneficial for Mamba, especially for small batch sizes, by adding more parallel computations across diagonals. This remains true with and without efficient CUDA kernels for Mamba (mamba_ssm).
> We also want to clarify that applying DB to other architectures would require model-specific grouped kernels and is outside the scope of this work.
>
> > W3. Reproducibility and accessibility
>
> The released code for the ARMT model with and without Diagonal Batching uses standard HuggingFace integrations, hence the model can be easily used with most toolchains or benchmarks. We also demonstrate it in the supplementary code by evaluating the model with Diagonal Batching on the BABILong task (the results are presented in Table 4 in Appendix D, the code for evaluation available in the supplementary materials).
> In supplementary materials we provide ARMT grouping mechanism for LLaMA and GPT models. For new architectures, one need to point on layers block and implement pre- and post-process functions (see usage_universal_executor.ipynb in supplement code with GPT and LLaMA examples).
>
> > W4. Evaluation bias
> > Q2. How does the method perform under concurrent multi-user loads
>
> We agree that the main text emphasizes latency, but the paper already evaluates numerical stability and memory: we report error accumulation between ARMT and ARMT+Diagonal Batching (showing small logit drift and no degradation on downstream tasks, Tables 3 and 4) and we show what ARMT can handle more load without OOM  in the appendix E.3 Table 4.
> We also expanded these evaluations to lengths up to 128k and added comparison not only on numerical stability of logits, but also on actually predicted tokens. Table.6 shows what error in tokens chosen by argmax do not exceed 2% on up to 128 segments. Regarding multi-request workloads, DB composes with standard batching. We focus on “one server - one prefill”, as it simplifies balancing and batching (you do not need chunked prefill or context based scheduling), while our DB can utilize GPU in that setup.

---

> > ### Author Response · Authors · 2025-11-22
> >
> > > W5. Overemphasis on speedups
> >
> > Our reported 3.3× speedup is the maximum observed across all configurations, not the typical value. After contacting the original ARMT authors [1], we confirmed that recent implementations typically use segment sizes of 512 and 1024 (while their original paper[1] used 128). Across all these segment sizes, our method delivers consistently large gains, as shown in Tables 1–2.
> > From a practical standpoint, we would like to emphasize that diagonal batching only requires changes at inference time. This allows us to dynamically select the most optimal strategy based on the hardware and usage patterns, such as segment size or regular sequential execution. Thank you for bringing this up, we will add this discussion.
> >
> > Most practical downstream tasks nowadays are solved with models, which can be runned under one GPU. Nevertheless, multi GPU setup is orthogonal to our research. Moreover, as workload per one GPU will lower, utilization problems (thus our Diagonal batching) will be even more important.
> >
> > > W6. Writing style and tone
> >
> > From our perspective, this level of technical details is needed for clarity and reproducibility. If there are particular sections that you find unnecessarily long or off-point, pointers to those would help us understand the concern more concretely.
> >
> > > Q3. Could you provide memory overhead statistics—does grouped GEMM increase peak memory usage?
> >
> > Thank you for raising this memory concern. In our implementation, each GroupedGEMM only allocates one output tensor of shape (group_size, segment_size, hidden_dim). For typical settings (e.g., group_size=20, segment_size=1024, hidden_dim=4096) in BF16, that’s 0.167 GB per layer (2010244096 * (2 bytes per bf16) bytes).
> >
> > Moreover, we leverage PyTorch’s caching allocator (via JIT, see code in attachment) to reuse the same buffer across layers, avoiding repeated large allocations or fragmentation. In practice, we observe no visible increase in VRAM usage on nvidia-smi when comparing Diagonal Batching to the original method. As allocators are widely used, this technique is also portable to any other implementation.
> >
> > > Q4. How significant are the numerical drifts beyond 64K tokens in real tasks?
> >
> > We evaluated the ARMT model with and without Diagonal Batching up to 128k tokens on the BABILong task, the results are presented in Table 4 in Appendix D. As one can see, the numerical drifts in the real task are negligible. We also calculated the drift in tokens, chosen with argmax for both models; the results are presented in Table 6 in Appendix D and show the same outcome.
> >
> > > Q5. Have you tried combining Diagonal Batching with quantized or sparsified models (e.g., AWQ or FlashDecoding)?
> >
> > We do not change model structure under ARMT wrappers, thus quantizations and sparsity could be applied as independent optimization. For that reason, we do not focus on them.
> >
> > References:
> >
> > [1] Rodkin, Ivan, et al. "Associative recurrent memory transformer." arXiv preprint arXiv:2407.04841 (2024). (https://arxiv.org/abs/2407.04841)

---

> > > ### Author Response · Authors · 2025-11-27
> > >
> > > Thank you very much for the detailed review!
> > > We have updated the paper and appendix to directly address your main concerns:
> > >
> > > 1. We clarify that our key contributions and show broader application of work.
> > >
> > > 2. We clarify HF-based integration and end-to-end BABILong code, add 128k-token drift and token-level error results, and provide memory overhead analysis. You may also be interested in empirical memory overhead we got from nvidia-smi, please see W2 answer for reviewer PbQn.
> > >
> > > 3. We clarify experiments setup concerns and label 3.3× as the maximum gain, while highlighting typical settings (segment 512/1024).
> > >
> > > Given these revisions and new experiments, we believe the paper now more fully addresses your concerns and strengthens the case for its contribution. If there are points you feel are still not fully resolved, we would be very grateful for further feedback. Otherwise, we would kindly ask you to consider updating your score.
> > >
> > > Best regards,
> > > The authors

---

### Official Review · Reviewer_PbQn · 2025-11-02

**Soundness:** 3
**Presentation:** 4
**Contribution:** 3
**Rating:** 8
**Confidence:** 2

**Summary:**

[Disclaimer that I worked on this field in 2023 and 2024 and have not kept up to date with the latest trends in 2025, so please take this review with a grain of salt.]

The paper introduces a new scheduling scheme for recurrent transformers, which the authors call "Diagonal Batching". The framing is as follows: RMTs have linear inference but are sequential and hence not easily parallelizable. Parallel RMTs localize recurrence within layers and eliminate all inter-layer memory flow. The issue is that PRMTs underutilize GPUs for single, long input requests. The paper introduces diagonal batching as a way to utilizes the GPUs better and hence improve latency. This is done by rearranging the layers and segments into a diagonal structure, where parallel operations can occur at the same time but across different layers.

The authors show that this provides better latency, especially in long token settings, though the results are less pronounced in the short token settings. They show that results hold across scales. The authors also provide an error analysis which shows that the error accumulation of the method is minimal.

**Strengths:**

- good exposition of background information and framing of the contributions of the paper
- proposed method seems simple and can be applied quite generally across different models. If indeed true, the method provides a lot of latency gains basically for free.
- method section seems quite complete. The authors go through both the high-level motivations but also outline the implementation details.
- experiments section seems complete as well. The authors show comparisons with different scales, sequence lengths, and baselines.

**Weaknesses:**

- The method is actually slower for sequence length 4096 and 8192.
- The tables and figures mainly show latency. I would have wanted to see the effect also on other metrics like memory or GPU utilization.
- The paper reported error accumulation numbers, but I would have wanted to also see the actual effect on the produced tokens, or maybe just verify that scores on some common benchmarks like MMLU remain the same.

**Questions:**

- How would these results change with smaller/larger GPUs? Every few years, new GPUs with larger memory and faster processing come out, so I'm curious if these would affect the results.
- Why is the error accumulation only upto 32k sequences? It seems like the main benefits of the method are more pronounced for longer sequences, so an error accumulation at these scales would be good to see as well.

---

> ### Author Response · Authors · 2025-11-23
>
> > W1. The method is actually slower for sequence length 4096 and 8192
>
> You are correct that Diagonal Batching can be slower at sequence lengths 4k and 8k in our current implementation. This is expected: the grouping logic has a fixed overhead, and for relatively short contexts the effective group size remains small, so this overhead is not amortized. In practice, we target the long-context regime, where standard quadratic attention is already unattractive and linear-time RMTs become relevant. As shown in Tables 1-2 and Figures 3-4, in this regime (>=16k tokens, or >=8k with shorter segments) the cost of grouping is fully hidden and Diagonal Batching provides clear gains. Importantly, Diagonal Batching is an inference-only change, so at deployment time we can dynamically choose the best strategy per request and hardware: plain sequential execution for short contexts, or Diagonal Batching for long contexts. This can ensure that we do not worsen latency when the method is not beneficial.
>
> > W2. The tables and figures mainly show latency ... other metrics like memory or GPU utilization.
>
> To be more transparent, we provide raw statistics taken from nvidia-smi under A100 80Gb Nvidia GPU. We did not include this in the appendix section because the measurements are not very precise.
>
> **Llama 1B bf16, segment = 512, mem = 16, memory_hidden = 64. Peak GPU util. and Peak mem - are maximum values displayed in nvidia-smi under repeated load with specified context.**
>
> | Method    | mem after init (MiB) | Context (tokens) | Peak GPU util. (%) | Peak mem (MiB) | Latency (s) |
> |----------|----------------------------|------------------|---------------|----------------|------------:|
> | ARMT     | 2951                       | 8,192            | 76%           | 4,465          |     0.573   |
> | ARMT     | 2951                       | 32,768           | 71%           | 8,721          |     2.300   |
> | ARMT     | 2951                       | 131,072          | 75%           | 23,056         |     9.140   |
> | ARMT+DB  | 3,293                      | 8,192            | 80%           | 6,689          |     0.259   |
> | ARMT+DB  | 3,293                      | 32,768           | 92%           | 6,809          |     0.900   |
> | ARMT+DB  | 3,293                      | 131,072          | 92%           | 8,255          |     3.480   |
>
> As can be seen, ARMT + DB does not introduce a large amount of memory overhead, nor does it grow significantly with an increased context size. Additionally, it can be observed that Diagonal Batching is still faster when the context is lower than N_layers * segment_size, even when full diagonal grouping can not be achieved. Even in these cases, processing the sequence start and tail is faster with grouping: group sizes are 1, 2, 3, …, N_layers, …, N_layers, …, 3, 2, 1.
>
> > W3. The paper reported error accumulation numbers, but I would have wanted to also see the actual effect on the produced tokens ...
>
> Since our goal is to optimize models for long sequences, we report our results on BABILong, a benchmark that tests a model’s ability to reason over long input sequences. The model is required to answer a question based on the provided context; the answer usually takes only 1-3 tokens. We found that the models’ outputs are the same with and without Diagonal Batching (Table 4 in Appendix D). To further validate the effect on the produced tokens, we calculated the relative difference in tokens, chosen with argmax (e. g. how many tokens with greedy generation differs in ARMT and ARMT with Diagonal Batching). The percentage values are added in Table 6 in Appendix D.
>
> > Q1. How would these results change with smaller/larger GPUs? ...
>
> Diagonal Batching is mostly a scheduling change, not a hardware-specific trick, so we expect the relative results to transfer across GPU generations.
>
> Our speedups come from exposing up to number of layers ​way parallelism and running grouped GEMMs/attention near the FLOP roofline. While memory stays tamed as we only need to allocate memory for that one requests’ state and output of one Grouped tensor (reuse across different layers is possible).
> On both A100 and H100, we already reach near-peak GEMM/Attention FLOPS once the group size is larger then 3–5 layers (shown in Figure 6-7 of Appendix sections D.1 and D.2).
> Future GPUs with 2–3 times more compute can still be saturated by grouping more layers/segments, especially in long-context RAG workloads, so the curves simply shift down in latency while utilization peak is still achieved. Additionally, the gap vs. sequential ARMT and full attention should remain similar or grow.
>
> > Q2. Why is the error accumulation only upto 32k sequences? ...
>
> Thank you for the valuable suggestion! We extended all of the main evaluations, such as error accumulation and BABILong evaluation, up to 128k sequences. The updated results are highlighted in the text with the red color. These results show the same consistent trend on the extended length.

---

> > ### Author Response · Authors · 2025-11-27
> >
> > Dear reviewer, PbQn!
> >
> > Thank you very much for the thoughtful review, and for the high overall assessment of our work. We appreciate your careful reading and the concrete suggestions for improvement.
> >
> > In response to your comments, we have:
> > 1. Clarify the behavior of diagonal batching on short sequences (4k–8k), explaining the fixed grouping overhead.
> >
> > 2. Show real GPU utilization and memory statistics from nvidia-smi on an A100 80GB, showing that Diagonal Batching introduces only modest memory overhead, achieves higher utilization, and remains faster even when the context is shorter than N_layers x segment_size.
> >
> > 3. Extended the analysis beyond BABILong accuracy (which remains unchanged) by reporting the relative percentage of differing argmax-generated tokens between ARMT and ARMT+DB (Table 6 in Appendix D).
> >
> > 4. Extended all main long-context evaluations, including error accumulation and BABILong, from 32k up to 128k tokens, confirming the same trends at longer scales.
> >
> > 5. Expanded the discussion on hardware dependence, explaining why we expect the relative speedups and utilization behavior to transfer to future GPU generations with higher compute.
> >
> > Thank you again for your constructive feedback - it helped us significantly improve the clarity and completeness of the paper. If you have additional questions, we will be glad to answer.

---

### Meta-Review · Area_Chair_SABN · 2026-01-06

**Summary:**

This paper provides an approach to for increasing parallelism in sequential models. The approach is simple, and involves arranging the computation in a 2D grid (across sequence length in x axis, depth on y axis), and then performing the computations diagonally (bottom left corner to top corner). This can enable greater parallelism as computations across layers can be shared.

Reviewers in general appreciated the simplicity of the method, which was supported by some latency gains with a particular architecture. Some concerns included lack of experiments on other architectures, as well as its limited scope to prefill inference.

While none of the reviewers raised this point, my main concern with the paper is novelty. This diagonal rearrangement idea goes back at least as far as the original seq2seq paper in 2014, which used essentially the same idea to parallelize computations in an LSTM (see section 3.5 of https://proceedings.neurips.cc/paper_files/paper/2014/file/5a18e133cbf9f257297f410bb7eca942-Paper.pdf). Given this, the main novelty here would be in applying this idea in a new setting, i.e., segment-level Transformer(like) models. But this seems too incremental.

**Reviewer Concerns:**

Major reviewer concerns included:
- Lack of experiments on other architectures
- Focus on pre-fill inference only
- Only showing latency numbers

The last point (i.e., only showing latency numbers) was somewhat addressed in the rebuttal, but the first two points were unaddressed.

**Reviewer Scores:**

Based on my guess:
Reviewer PbQn: kept score
Reviewer XxMs: kept score
Reviewer qKhy: kept score

---

### Decision · Program_Chairs · 2026-01-26

Reject